# Congenital Zika Syndrome—Assessing the Need for a Family Support Programme in Brazil

**DOI:** 10.3390/ijerph17103559

**Published:** 2020-05-19

**Authors:** Antony Duttine, Tracey Smythe, Míriam Ribiero Calheiro de Sá, Silvia Ferrite, Maria Zuurmond, Maria Elisabeth Moreira, Anna Collins, Kate Milner, Hannah Kuper

**Affiliations:** 1International Centre for Evidence on Disability, Department of Clinical Research, Faculty of Infectious and Tropical Diseases, London School of Hygiene & Tropical Medicine, London WC1E 7HT, UK; tracey.smythe@lshtm.ac.uk (T.S.); maria.zuurmond@lshtm.ac.uk (M.Z.); annalucy0910@gmail.com (A.C.); hannah.kuper@lshtm.ac.uk (H.K.); 2Instituto Nacional de Saúde da Mulher, da Criança e do Adolescente Fernandes Figueira, Rio de Janeiro 22250-020, Brazil; calheirosa@uol.com.br (M.R.C.d.S.); bebethiff@gmail.com (M.E.M.); 3Department of Speech and Hearing Sciences, Institute of Health Sciences, Federal University of Bahia, Salvador 40110-902, Brazil; ferrite@ufba.br; 4Murdoch Children’s Research Institute and Department of Paediatrics, University of Melbourne, Parkville, Victoria 3052, Australia; kate.m.milner@rch.org.au

**Keywords:** congenital zika syndrome, Zika, family support, Brazil, cerebral palsy, community programme

## Abstract

The Zika outbreak in Brazil caused congenital impairments and developmental delays, or Congenital Zika Syndrome (CZS). We sought to ascertain whether a family support programme was needed and, if so, could be adapted from the Getting to Know Cerebral Palsy programme (GTKCP) designed for children with cerebral palsy (CP). We conducted a systematic review of the needs of families of children with CZS or CP in low- and middle-income countries and reviewed the findings of the Social and Economic Impact of Zika study. We undertook a scoping visit to three facilities offering services to children with CZS in Brazil to understand potential utility and adaptability of GTKCP. The literature review showed that caregivers of children with CZS experience challenges in mental health, healthcare access, and quality of life, consistent with the CP literature. The scoping visits demonstrated that most support provided to families was medically orientated and while informal support networks were established, these lacked structure. Caregivers and practitioners expressed an eagerness for more structure community-based family support programmes. A support programme for families of children with CZS in Brazil appeared relevant and needed, and may fill an important gap in the Zika response.

## 1. Introduction

Since the peak of the Zika epidemic in 2015–2016, the number of new cases of Zika infection and of confirmed Congenital Zika Syndrome (CZS) has gradually declined across the Americas [1], although the virus is now considered endemic to the region [2]. Brazil was the most heavily impacted country, accounting for 47.9% of total cases between 2016 and the end of 2019 [3]. The Ministry of Health in Brazil reports that since 2015, there have been 3474 confirmed and 743 probable cases of CZS with a further 2659 cases under investigation [4]. There are likely to be many more cases that have not been designated as caused by Zika, given the emerging evidence of more mild and later onset impairments and the lack of a reliable retrospective test for Zika. 

CZS was defined by Moore et al. [5] as a syndrome of congenital anomalies associated with Zika virus (ZIKV) infection during pregnancy including; severe microcephaly with a partially collapsed skull; thin cerebral cortices with subcortical calcifications; eye anomalies, including macular scarring and focal pigmentary retinal mottling; congenital contractures or a limited range of joint motion; marked hypertonia; and symptoms of extrapyramidal involvement.

While many of these characteristics are common features of congenital central nervous system infections, it was the epidemic of microcephaly and children with severe neurodevelopmental sequelae that initially raised alarm during the Zika epidemic [5]. Subsequently, during the course of the Zika crisis, there was increasing recognition of the broader spectrum of anomalies occurring in children with CZS [6]. For example, while microcephaly was common, it was by no means always present for cases of CZS, and while ophthalmologic manifestations often co-occurred with other neurological features, there were case reports of these occurring in isolation [6]. 

Emerging evidence on CZS suggested that affected children appeared to have motor abnormalities consistent with internationally accepted definition of cerebral palsy (CP): “A group of permanent disorders of the development of movement and posture, causing activity limitation, that are attributed to non-progressive disturbances that occurred in the developing fetal or infant brain” [7]. The presence of microcephaly in CZS is strongly associated with severe neurological disabilities, such as hearing problems, epilepsy, and learning disabilities, which are also common in children with CP [8,9]. Indeed, in situations such as the Zika epidemic, when a specific cause for the occurrence of cerebral palsy is known, there is international consensus that the term cerebral palsy be used in addition to naming the underlying cause [10].

The focus of research and response to Zika in the Americas has fallen heavily on understanding the pathophysiology of the virus, prevention of the spread of the virus, and development of a vaccine and treatments [11]. In contrast, relatively little focus or specific research considered how to meet the needs of children born with CZS, or the impact of CZS experienced by their families, and they are likely to experience widespread unmet needs.

Interventions are therefore needed to meet the broader needs of children with CZS. A number of family-based support programmes have been established for families of children with neurodevelopmental disability in low- and middle-income contexts to respond to the unmet needs experienced by these families [12,13]. One such programme is Getting to Know Cerebral Palsy (GTKCP), developed to educate and empower caregivers in the care of their child [14]. GTKCP is a community based participatory programme for caregivers in a support group setting, and has been shown to improve caregiver quality of life and knowledge and confidence in caring for a child [15,16]. By targeting the caregivers, the programme aims to have an impact on the long-term health, wellbeing, and participation of children with cerebral palsy. An Early Intervention Programme (EIP) has also been developed from GTKCP to address the needs of younger infants [17]. Considering the commonalities of CZS and CP, it may be plausible to use existing CP programmes as the basis for new interventions. Therefore, the aim of the current study was to ascertain whether a similarly structured family support programme to GTKCP was (a) needed and (b) relevant for the post-Zika Brazilian context.

Specific questions that this study sought to answer were: a)What are the needs of families of children with CZS (or related conditions) in Brazil, and are they being met by the existing support services?;b)Would a family support programme be potentially useful in the post-Zika context in Brazil?;c)Are the similarities between CP and CZS sufficient to suggest that GTCKP/EIP could be used as a basis for a Brazil family support intervention?

## 2. Materials and Methods

### 2.1. Data Extraction

Data were extracted to a custom-made spreadsheet in Microsoft Excel. We did not perform a meta-analysis, as the data were highly heterogeneous and included qualitative and quantitative data. 

### 2.2. Ethical Approval

Ethics approval was acquired in Brazil (IFF/FIOCRUZ-RJ/MS 2.183.547) and the UK (LSHTM Ethics number 13608).

### 2.3. Methods

The methods involved 3 processes:

#### 2.3.1. Systematic review on unmet needs of families of children with CZS and CP

A systematic search of the literature was performed in June 2017 to identify articles that considered the unmet needs of families of children with CZS [18]. The review was led by one researcher (AC), with a second researcher (AD) acting in a supervisory capacity. We followed the Preferred Reporting Items for Systematic Reviews and Meta-Analyses (PRISMA) Statement [19] in the conduct of this systematic review. We searched CINAHL Plus, EMBASE, MEDLINE, Psychinfo, and PubMed, and search terms are included in Appendix A1. Inclusion criteria were: Articles related to the needs or wellbeing of families with children with CZS, published from 2000–2017 in peer reviewed journals, full text available in English. No exclusion criteria were set in terms of study design, due to the lack of research into the topic at the time. 

Few articles were identified as being relevant. Therefore, a second search was conducted, expanding search terms to include articles related to CP. Inclusion criteria were the same as above, however this time related to the needs or wellbeing of mothers and families with infants or children with CP and only research set in lower middle- or upper middle-income countries was considered. 

In both parts of the search, articles were first reviewed for relevance by title, followed by abstracts and full texts by one researcher (AC). Duplicates and articles that did not meet inclusion criteria were excluded. Identified texts were confirmed for relevance by a second researcher (AD). Once identified, references were saved and managed using Mendeley Web and Mendeley Desktop. A PRISMA flow chart is included in Appendix B.

#### 2.3.2. Findings from the Social and Economic Impact of Zika Study

Emerging themes that were being raised as part of a parallel study, *The Social and Economic Impacts of CZS on Families and Caregivers* [20] were reviewed. This was a mixed methods study, conducted in Recife and Rio de Janeiro. It included in-depth qualitative interviews in each setting with approximately 30 families of children with CZS and 10–12 healthcare providers, as well as a case-control study of 163 children with CZS and 324 unaffected controls. Through both approaches, information was collected on economic, mental health, and social impacts, using standardised tools in the quantitative component (e.g., Depression Anxiety Stress Scales (DASS) for depression, anxiety and depression, and Medical Outcomes Study Social Support Scale for social support). An additional statistical analysis was conducted using data from this study, comparing the results of the Pediatric Quality of Life Inventory (PedsQL) Family Impact Module between a subset of participants from the study—155 mothers of children with CZS and 47 mothers of unaffected children—in order to assess the broader impacts of CZS on the quality of life of families. For logistical reasons, the PedsQL data were not collected from all participants. The PedsQL Family Impact Module is a questionnaire that measures parent self-reported physical, emotional, social, and cognitive functioning, communication, and worry. The module also measures parent-reported family daily activities and family relationships. It is scored on a 5-point Likert scale where 0 is “never” and 4 is “almost always”. The results are then transformed to a 0–100 scale to enable/allow scoring and data analysis. We used the PedsQL Family Impact Module to compare the parent self-reported physical, emotional, social, and cognitive functioning, and communication and worry between mothers of children with CZS and mothers of children with unaffected children. 

#### 2.3.3. Scoping visit in Brazil

In April 2017, 3 researchers (AD, MZ, HK) undertook a week-long visit to Brazil. The researchers visited a range of facilities offering services to children with CZS and their families. The sites were identified and selected by the local research partners in Brazil (SF, MS, EM) and included, a tertiary facility in Rio de Janeiro (Instituto Fernandes Figueira (IFF)), which offers clinical services including habilitation and psychosocial support in a hospital based setting in Central Rio de Janeiro; the Altino Ventura Foundation, an NGO in Recife providing support and care to families, including group programmes, in a hospital-based setting; and Associação aBRAÇO a Microcefalia, a parent support programme in Salvador, Bahia which offered twice monthly meetings of carers, including both formal lectures and social activities, as well as therapeutic support and donations (e.g., nappies/diapers). At each site, the researchers consulted with caregivers of affected children, and health care professionals (doctors, psychologists, physiotherapists, occupational therapists, social workers, speech therapists, and lactation specialists). Caregivers (n = 7) were consulted about the services offered, the main perceived barriers and gaps, and level of interest in a formal parent support programme. Consultations with healthcare professionals (n = 12) included mapping the flow of service delivery to meet child and family support needs within the existing structures of Brazil, in order to better understand and contextualise how services are currently delivered to families of children with CZS.

## 3. Results

### 3.1. Findings from the Literature Review of Families and Caregivers of Children with CZS and CP

Only seven eligible papers were identified that assessed the needs of children with CZS [21,22,23,24,25,26,27], and 31 eligible papers focussed on the needs of children with CP [18]. Table 1 and Table 2 below summarise the findings.

Studies focussing on psychosocial aspects of caring for a child with CZS or CP found higher levels of anxiety and depression and poorer Quality of Life (QoL) scores in primary caregivers of children with these conditions, which was usually the mother. CZS literature was still emerging, but one study provided specific information on the impact on the psychosocial domain of caregivers with CZS had interviewed mothers within 24 h of birth of a child [21]. Lack of sleep of parents of children with CZS due to severe cerebral irritation noted in these children may compound psychosocial distress [27].

Expanding to literature on CP, we noted findings around financial hardships, difficulties with transport and services, and stigma. These challenges were associated with a significant impact on the caregivers’ psychosocial wellbeing. Difficulties were also reported by caregivers of children with CP in terms of lack of access to services, in particular on account of distance, cost and lack of availability. The information needs of parents were cited in one study as being greater than financial or other support needs [28]. Of the different types of information gaps, information about the ‘child’s condition’ and information about the ‘institutions that the child can benefit from’ were the two most frequently reported [28].

### 3.2. Findings from Social and Economic Study

The social and economic impacts of the CZS study also highlighted the needs of parents of children affected by CZS. The quantitative data showed that mothers of children with CZS were more likely to experience depression, anxiety, and stress than mothers of unaffected children [29]. Mothers of children with CZS reporting low social support were particularly likely to experience depression, anxiety, and stress, indicating that social support may buffer adverse mental health effects. 

This study also showed through qualitative and quantitative data that affected children had very high health care needs, and had to make frequent visits to services to attend to specific conditions related to CZS (neurology appointments, physiotherapy etc.), co-morbidities (e.g., chest infection, epileptic seizure) and routine health care needs (e.g., vaccines) [30]. Services were often far away, fragmented, and uncoordinated. As a consequence, healthcare professionals felt that it was difficult to adequately meet the holistic needs of these children and their families. Visits by families to therapy and medical appointments focussed almost exclusively on the therapeutic or medical interventions and parents felt they had little opportunity to discuss their own needs. Parents also reported issues of distrust with healthcare professionals based on difficulties in communication with the health care provider. 

Some identified gaps in services included provision for children with less severe developmental disability and delays. Concerns were expressed by several health professionals about children with mild CZS being lost from the system, either due to parents not believing that the impairments warranted intervention or because the systems in place had been established for more severe cases. In addition, families reported high household expenditures to meet the healthcare needs of their child. This impact was particularly difficult since the families were on average poorer than families of a child with a disability (paper in submission).

The PedsQL analyses conducted for this paper showed that mothers of children with CZS had worse quality of life scores across all domains (Table 3). These differences reached statistical significance in relation to problems with communication and problems with worrying, showing that these are important needs that should be addressed. 

### 3.3. Findings from Scoping Visit

Our mapping of services indicated that the response to CZS had a largely medical/therapy-based focus. The main structure in Brazil is the provision of interventions at specialised tertiary level health centres, which tended to be based in large urban settings (e.g., Rio, Salvador, Recife). The Brazilian Unified Health System, Sistema Único de Saúde, has an extensive health network reaching out to primary level settings. However, rehabilitation teams are not always available at the primary level, either due to lack of specialised rehabilitation staff or services not being developed. Those staff that are based at primary level may be more likely to be generalists, and lack the specialised paediatric knowledge and experience required to meet the needs of many of the children with CZS. 

Other forms of support were available for some families in some settings. Similar grassroot family-support initiatives to those seen at Associação aBRAÇO a Microcefalia existed in other settings, and it was reported by both caregivers and healthcare providers that families often made informal social networks to be connected with others. These initiatives varied in focus and structure. Some groups had more of a focus on advocacy and promoting children’s rights, rather than on caregiver education and support. Additionally, mothers almost universally reported being part of WhatsApp groups with other carers, which provided some social and emotional support, but was unstructured and on an ad hoc basis.

Health professionals reported that the concept and approach of GTKCP was highly relevant for the situation being faced by many Brazilian families. Most caregivers said that having support groups would be acceptable to them and welcomed the idea of having an opportunity to learn and share from one another. However, practical considerations were also raised with respect to the parent support groups. The security situation, particularly in Rio de Janeiro, was a concern, because levels of urban violence meant that the logistics of safely planning community-based interventions were more complicated. There was also awareness that additional programmes should complement clinical services and ideally be integrated with caregiver networks already in place.

## 4. Discussion

This paper aimed to ascertain whether a structured family support programme to GTKCP was needed and relevant for the post-Zika Brazilian context. Specifically, this mixed-method study generated evidence to respond to the three following questions posed, as follows:a)What are the needs of families of children with CZS (or related conditions) in Brazil, and are they being met by the existing support services?

We found a need for caregivers to receive a higher level of informational, psychosocial, and emotional support than was currently available. Health and specialist medical needs, including rehabilitation, were largely available and being accessed by families. However, meeting the health care and specialist needs of children was onerous, especially given the need to travel long distances to access the relevant services. This finding is echoed in recent publications, which have described several family impacts, notably isolation, stress, lack of access to services, and powerlessness [31] and the importance of a holistic approach to meet the broad needs of children with neurodevelopmental disabilities and caregivers [32]. 

b)Would a family support programme be potentially useful in the post Zika context in Brazil?

A family support programme may be useful in the post Zika context in Brazil as a complement to clinical services and existing caregiver groups and networks. Both caregivers and health professionals agreed that a support programme could be an important adjunct to the existing services and fill gaps in the existing support mechanism, such as a focus on mental health of caregivers and holistic needs of children. The needs of caregivers of children with less severe or later onset impairments are an important group to consider. 

Parenting programmes have been shown to have a positive impact on self-efficacy for parents of children with developmental disabilities [33]. Targeting parent and caregiver skills and behaviour can have an important foundational impact on child health and wellbeing. If a family support programme in Brazil can have some similar impacts on children and caregivers as GTKCP has shown [15,16,34], these foundations can potentially impact more long-term health and wellbeing outcomes of children with CZS across the life-course.

c)Are the similarities between CP and CZS sufficient to suggest that GTCKP/EIP could be used as a basis for a Brazil family support intervention?

The evidence on CZS that was emerging at the time of this study highlighted the similarities of the physical presentation of CZS to those of CP. More recent literature has further described the overlapping between CZS and CP [35,36,37]. The review of the literature and findings of the social and economic impact study also suggested that there are similarities between CP and CZS, particularly in terms of needs and unmet needs of the caregiver. This led us to suggest that the GTKCP/EIP programmes had potential utility for being a basis of a programme in Brazil. However, it was also clear that adaptations to the existing programmes (GTKCP and EIP) to the context of Zika and Brazil would be required, to cater to the specific circumstances in Brazil and to address the mental health impacts on caregivers.

The needs of children with developmental disabilities will change over their life course, and consequently, family support programmes need to be adapted for different age groups. In the early years, programmes may need to focus on maximising development of the child, and supporting carers in looking after the child. As children reach the age of 5 or more, attention needs to be given to helping carers support their child’s inclusion in education. In the next stage, as children with developmental disabilities reach adulthood, focus of programmes should shift towards supporting independent living, employment, and maintaining health and function. Throughout the life course, programmes should address social inclusion and supporting carers, especially with respect to mental health.

Strengths of the mixed methods approach of the study include obtaining data from a range of sources and ensuring inclusion of different perspectives to create an overview of the needs of caregivers. For instance, data from the parallel study on social and economic impacts of Zika provided information that helped to confirm and elaborate findings from the scoping visit. Limitations include the fact that the Brazil scoping visit was a rapid, pragmatic stakeholder assessment in the setting of the epidemic, rather than a detailed qualitative evaluation, and a limited number of facilities were visited. Other limitations were that double screening was not undertaken within the systematic review, and the PedsQL was not collected on the full sample of cases and controls, which may be introduced biases. Finally, data on CZS was newly emerging when the literature review was undertaken, therefore giving a narrow range of useful information specific to caregiver needs relating to CZS.

## 5. Conclusions

A family support programme could potentially fill a gap in the range of services provided in Brazil in the wake of the Zika outbreak and could address unmet holistic needs of families of children affected by CZS. The literature review coupled with site specific needs assessment demonstrated an important gap in support for children and families affected by CZS. Further implementation research regarding contextual design and adaptation of psychosocial support programmes for caregivers of children with CZS and other neurodevelopmental disabilities at community level in low- and middle- income countries is urgently needed. However, given the similarities of CP and CZS, there could be justification to use GTCKP as a basis for a Brazil family support intervention if cultural and practical adaptations to the existing programme are made.

## Figures and Tables

**Table 1 ijerph-17-03559-t001:** Overview of findings from Congenital Zika Syndrome articles.

Article	Country	Measures Used	Overall Findings/Topics
Anxiety, depression, and quality of life in mothers of newborns with microcephaly and presumed congenital Zika virus infection [21]	Brazil	World Health Organisation Quality of Life-BREF (WHOQoL-BREF)	Lower scores in psychosocial domain of WHOQoL-BREF of women with babies with microcephaly in first 24 h after birth.
Babies with microcephaly in Brazil are struggling to access care [22]	Brazil	Anecdotal evidence	Struggle of families to access care, transportation, investigation, and medication. Financial cost of bringing up an infant with congenital Zika syndrome.
Congenital Zika virus infection: A developmental- behavioural perspective [23]	Brazil	Anecdotal evidence and recommendations	Stigma surrounding congenital zika syndrome in Brazil. Broad range of outcomes and potential interventions needed.
Engaging human rights in the response to the evolving Zika virus epidemic [24]	Brazil	Relationship between human rights principles and Zika response with relation to discrimination, participation, accountability of Brazilian health system, equity	Health system may need to divert resources to areas of greatest need, given that Zika was concentrated in areas that may have less health providers. Need to address structural and social determinants of health.
Integrated reproductive health: The Zika virus [25]	Brazil	Anecdotal evidence	Psychological impact on women and need for support and communication. Social inequities within Brazil.
Infants with congenital zika virus infection: A new challenge for early intervention professionals [27]	Brazil	Recent literature and recommendations	Social stigma and media attention may affect parents’ psychological wellbeing. Poor sleep patterns of infants may contribute to poor emotional health of parents. Parents may need education and explanation of child’s condition. Adequate psychosocial services will be necessary, as well as possibly respite opportunities. Consultation with lactation specialists may be useful.
Brazil struggles to cope with zika epidemic [26]	Brazil	Anecdotal evidence	Highlights lack of available finances and services in the Brazilian health system.

**Table 2 ijerph-17-03559-t002:** Overview of findings of CP studies [18].

Article	Country	Scale or Questionnaire Used	Main Findings
Understanding the lives of caregivers of children with cerebral palsy in rural Bangladesh: Use of mixed methods	Bangladesh	PedsQL Family Impact Questionnaire	Lower quality of life in all domains of PedsQL in families of children with CP (*p* < 0.001). Parents experienced fatigue, stigma, lack of social support.
Assessment of family environment and needs of families who have children with cerebral palsy	Turkey	Family Needs Score (FNS) and Family Environment Score (FES)	Vast majority (91.8%) of primary caregivers were mothers. More families cited information needs (84.3%) than support or financial needs. Many families had assistance from elders.
An investigation of parents’ problems according to motor functional level of children with cerebral palsy	Turkey	Author written questionnaire	Families with children with more severe CP had more problems than those with mild CP (no *p*-value given). Major difficulties were economic, lack of health services, and communication. Many families had assistance from elders.
Comparative quality of life of Nigerian caregivers of children with cerebral palsy	Nigeria	World Health Organisation Quality of Life score (WHOQoL-BREF), Gross Motor Functional Classification System (GMFCS)	Caregivers of children with CP have a lower quality of life than those without children with CP (*p* = 0.003). Quality of life scores improved over time as children’s motor function improved, suggesting that early intervention and therapy may help with caregiver’s quality of life long term. No significant correlation between child’s GMFCS and severity of depression (*p* = 0.339).
Depression in mothers of children with cerebral palsy and its relation to severity and type of cerebral palsy	Iran	Beck Depression Inventory-II (BDI-II), GMFCS	Greater risk of mothers caring for children with CP having depression (*p* = 0.003). No significant correlation between the GMFCS and severity of depression.
Depression and anxiety levels in mothers of children with cerebral palsy: A controlled study	Turkey	Beck Depression Inventory (BDI), Beck Anxiety Inventory (BAI)	Higher levels of depression and anxiety in mothers of children with CP (*p* = <0.001). Statistically significant difference in effect of speech defects and higher GMFCS score on mothers’ depression (*p* < 0.05 with 95% CI) based on logistic regression.
Depression in parents of children with cerebral palsy in Bosnia and Herzegovina	Bosnia and Herzegovina	Zung self-evaluated method for depression	No significant difference in levels of depression between mothers and fathers of children with CP, and mothers of healthy controls (*p* = 0.09).
Factors associated with caregiver burden among caregivers of children with cerebral palsy in Sri Lanka	Sri Lanka	WHOQoL-BREF ‘Caregiver Difficulties Scale’ (CDS)	Majority of caregivers (97%) were mothers. Majority of caregivers were from a rural area and low socioeconomic background (72% and 70% respectively). Living in a rural area (*p* = 0.001), having a lower income (*p* < 0.023), male sex of the child (*p* = 0.017), and more significant functional impairment of child (*p* < 0.001) were associated with a higher caregiver burden in multivariate analysis. Social support was associated with a lower caregiver burden (*p* < 0.001).
Functional priorities reported by parents of children with cerebral palsy	Brazil	Questionnaire	In all age groups, ‘personal care’ was the highest rated functional goal by parents (42.99%–52.38%). In 3–6 year olds, play was second highest rated (20.56%), in 7–10 year olds and in 11–16 year olds, school was the second highest rated (23.16% and 22.22% respectively).
Higher Levels of Caregiver Strain Perceived by Indian Mothers of Children and Young Adults with Cerebral Palsy Who have Limited Self-Mobility	India	Caregiver Strain Index (CS)	Caregivers of children with higher scores on GMFCS had higher levels of caregiver strain (*p* < 0.01).
Life quality among mothers of children with cerebral palsy living in Armenia	Armenia	BDI-II and Norakidze’s modification of Taylor manifest anxiety scale	High levels of depression (74%) and anxiety (95%) in mothers of children with CP. Mothers with lower level of education had higher rates of anxiety.
Mental health and quality of life of caregivers of individuals with cerebral palsy in a community-based rehabilitation programme in rural Karnataka	India	General Health Questionnaire (GHQ) WHOQoL-BREF	Majority (87%) of caregivers were mothers. No statistically significant difference in GHQ-28 score in relation to functional status of child. No statistically significant difference in children’s needs in relation to mothers’ mental health score.
Coping with stress and adaptation in mothers of children with cerebral palsy	Serbia	Family Crisis Oriented Personal Evaluation Scale (F-COPES)	No difference in methods of coping between urban and rural mothers, reframing was the strategy most commonly used. Only statistically significant difference in methods of coping in relation to severity of child’s functional impairment was use of institutions in more severe impairment.
Fatigue in the mothers of children with cerebral palsy	Turkey	Fatigue Symptom Inventory (FSI), Beck Depression Scale (BDS), and Nottingham Health Profile (NHP)	Mothers of children with CP scored higher in all groups of FSI (*p* < 0.00001). Mothers of children with CP had higher scores on BDS (*p* < 0.00001). Mothers of children with CP had higher scores on BDS (*p* < 0.026 or less in all domains). No impact of GMFCS on outcomes in mother when regression analysis applied. Fatigue correlated with higher NHP and BDS scores.
Predictors of stress in mothers of children with cerebral palsy in Bangladesh	Bangladesh	Judson Scale, Family Support Index (FSI)	Higher levels of stress in mothers living in rural areas (*p* = 0.02). Higher levels of household income associated with lower levels of stress (*p* = 0.02). Level of child’s functional impairment not associate with higher levels of stress. Child’s behavioural issues (including sleep, bet wetting, hyperactivity) associated with a higher level of stress (goodness of fit 75.46%).
Psychological distress and perceived support among Jordanian parents living with a child with cerebral palsy: A cross sectional study	Jordan	GMFCS, Perceived Stress Scale (PSS), BDI, Strengths and Difficulties Questionnaire (SDQ), and Multidimensional Scale of Perceived Social Support (MSPSS)	Many parents of children with CP have perceived levels of stress. Parents of children with higher GMFCS had higher levels of stress (*p* = 0.03). Parents of children with more behavioural issues had higher levels of perceived stress. Parents with lower social supports had higher levels of stress (*p* < 0.0005).
Psychological adversities and depression in mothers of children with cerebral palsy in Nigeria	Nigeria	Psychosocial Adversity Scale (PAS) and Patient Health Questionnaire (PHQ)	Additional psychosocial stressors associated with depression (all except unemployment and mother’s education). Majority of mothers (89%) had some degree of depression.
Quality of life in mothers of children with cerebral palsy: The role of children’s gross motor function	Iran	Short Form Health Survey (SF-36), GMFCS	Mothers of children with better GMFCS had better QoL scores. When compared with general population mean, mothers of children with CP has statistically significant lower scores in all QoL domains.
Quality of life in parents/caretakers of children with cerebral palsy in Kampong Cham, Cambodia	Cambodia	Comprehensive Quality of Life Scale (ComQOL-A5) scores	Lowest scoring QoL domains were health, emotional wellbeing, and material well-being.
Social support provided to caregivers of children with cerebral palsy	Brazil	Sarason’s Social Support Questionnaire (SSQ)	Majority of caregivers (88%) are mothers. Husband, mother, and brother are those cited most frequently as sources of social support.
The effect of having a children with cerebral palsy on quality of life, burn-out, depression and anxiety scores: A comparative study	Turkey	WHOQoL-BREF, GMFCS	Higher levels of depression in CP group compared to control group (58.0% vs. 46.7%). Higher levels of anxiety in CP group compared to control group (71.4% vs. 51.7%). Highest scores in WHOQoL were in domains of physical, psychosocial, and environment. Correlation between higher GMFCS and higher total WHOQoL and BDI scores (*p* = 0.04 and 0.01 respectively).
Quality of life and anticipatory grieving among parents living with a child with cerebral palsy	Jordan	Marwitand Meuser Caregiver Inventory Quality of Life Index	62.7% reported stress, 78.3% reported drastic life changes, 71.0% reported anxiety, 73.4% reported excellent family support. Personal sacrifice burden score highest. Negative correlation between anticipatory grief and QoL scores (*p* < 0.0005).
Psychosocial impact of caring for children with cerebral palsy on the family in a developing country	Nigeria	Impact on Family Scale (IFS) and GMFCS	Majority of caregivers (80.3%) were mothers. Although 46.2% of CP children had speech impairments, only 2.6% received speech therapy. Correlation between higher GMFCS and higher IOF scores, but not statistically significant (*p* = 0.16). Higher IOF scores in families of children with CP (*p* = 0.000).
Psychosocial challenges for parents of children with cerebral palsy: A qualitative study	Iran	Semi-structured interview	Lack of financial support, transportation, medical services. Sense of guilt, stigma. Lack of social support.
Investigation of quality of life in mothers of children with cerebral palsy in Iran: Association with socio-economic status, marital satisfaction and fatigue	Iran	WHOQoL-BREF, Socioeconomic Status Questionnaire (SES), Index of Marital Satisfaction (IMS) and Fatigue Severity Scale-Persian (FSS-P)	Mothers in CP group has lower SES categories. Mothers in CP group had higher fatigue levels (*p* < 0.001) and higher marital dissatisfaction (*p* < 0.001). Mothers in CP group had lower QoL scores in all domains (*p* < 0.001).
Frequency and severity of depression in mothers of cerebral palsy children	Pakistan	Siddiqui -Shah Depression Scale (SSDS)	50.62% of mothers had depression.
Experiences shared through the interviews from fifteen mothers of children with cerebral palsy, sexuality and disability	Turkey	Semi-structured questionnaire	Majority of caregivers were mothers, often blamed for child’s condition. Out of 12 who had other children, 3 reported difficulties in sibling relationships. All mothers reported financial difficulties. Lack of suitable support for child’s education. Concerns for child’s future.
An evaluation of quality of life of mothers of children with cerebral palsy	Turkey	Turkish version of SF-36	Negative correlation between SF-36 QoL scores and GMFCS; significant in domains of role physical (*p* = 0.001), bodily pain (*p* = 0.023), general health (*p* = 0.031), social functioning (*p* = 0.0320, role emotional (*p* = 0.003), and mental health (*p* = 0.004). Statistically significant difference between mothers of children with CP and controls in domains of mental health (*p* = 0.002), social functioning (*p* = 0.002), general health (*p* = 0.001), bodily pain (*p* = 0.005), and role physical (*p* = 0.008).
Assessment of the quality of life of mothers of children with cerebral palsy (primary caregivers)	Turkey	Nottingham Health Profile-1, BDI, BAI, GMFCS	Higher NHP score in mothers of children with CP in sleep, energy, social isolation (*p* = 0.000), pain (*p* = 0.007), physical activity (*p* = 0.004), and emotional reactions (*p* = 0.001). BDI scores higher in mothers of children with CP (*p* = 0.000). 78.2% of mothers of children with CP had depression compared with 21.7% in control group.
Coping strategies and resolution in mothers of children with cerebral palsy	Serbia	Reaction to Diagnosis Interview (RDI) and classification system used and modified version of F-COPES and Functional Status II (FS-II)	59% mothers remained unresolved. Reframing was the coping strategy used most, followed by passive appraisal. No difference between resolution and non-resolution depending on coping strategy. Mothers with children with better functional status who utilised institutional support had better resolution.
Depression in mothers of children with cerebral palsy and other related factors in Turkey: A controlled study	Turkey	BDI, GMFCS	More mothers in the CP group (61.2%) were depressed compared with control group (36%). Depression did not vary depending on CP type. Depression correlated with speech deficits (*p* = 0.036). No correlation between GMFCS level I, II, III and groups IV, V, and depression (*p* = 0.260). Higher BDI score correlated with lower household income (r = −0.384, *p* = 0.007).

**Table 3 ijerph-17-03559-t003:** The results of the PedsQL Family Impact Module comparing mothers of children with CZS to mothers of unaffected children.

Dimensions of PedsQL	Mothers of children with CZS (n = 155)	Mothers of Children with Unaffected Children (n = 47)	*p*-Value (*t*-Test)
Physical Functioning	53.6 (1.8)	54.6 (3.5)	0.39
Emotional Functioning	57.6 (1.9)	62.1 (3.3)	0.13
Social Functioning	56.7 (2.3)	61.6 (3.9)	0.15
Cognitive Functioning	60.5 (2.0)	66.1 (3.7)	0.09
Communication	58.9 (2.4)	71.6 (4.5)	0.006
Worry	33.6 (1.4)	38.9 (3.3)	0.04
Daily Activities	35.2 (2.1)	38.3 (4.2)	0.24
Family Relationships	60.5 (2.2)	58.0 (4.6)	0.70
Total	52.5 (1.3)	56.4 (2.7)	0.08

Note: Mean scores out of 100, with a higher number equating to higher reported quality of life; Standard deviation noted in parentheses.

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
