# Peer review of "Congenital Zika Syndrome—Assessing the Need for a Family Support Programme in Brazil"

_ijerph, 2020, doi:10.3390/ijerph17103559_

Round 1

Reviewer 1 Report

Reviewer:  The manuscript from Antony Duttine et al. describes the need for a family support programme regarding Zika virus infections.              

The authors undertook literature studies and visited care facilities on location. The authors came to the conclusion that an improved “Getting to Know Cerebral Palsy Programme” could be a working base for family support in Brazil.

Minor points:

Reviewer: In the introduction the authors formulate three clear questions. I suggest, following that structure more accurately in the discussion section to gain a better understanding of the main topics. I invite the authors to do so, but it is not absolutely necessary.

Author Response

Dear reviewer, 

Many thanks for your feedback and comments. With regards specific comments, we have made the following:

Your Comment: In the introduction the authors formulate three clear questions. I suggest, following that structure more accurately in the discussion section to gain a better understanding of the main topics. I invite the authors to do so, but it is not absolutely necessary.

Our Response: We have redeveloped the discussion to better reflect responses to the questions posed in the introduction, integrating the three questions posed.

Reviewer 2 Report

Comments for the article, “Congenital Zika Syndrome – assessing the need for a family support program in Brazil”

The article addresses a likely gap existing in third world countries. The Zika outbreak has adversely impacted the lives of many families with children born of Congenital Zika Syndrome (CZS) encompassing a broad range of congenital and developmental abnormalities. The article highlights that the availability of emotional and educational support for primary caregivers of the children suffering from CZS in the form of therapy and counselling is limited and its requirement maybe underrecognized and underrated, especially in low-middle income families. The authors point out that many features of this syndrome fit the description of cerebral Palsy and that these similarities can be used to exploit the existing support programs for cerebral palsy and tailor them to fit to CZS. The main aim of this article is to identify if the need of such support groups to Zika affected families is pragmatic and if yes, can the support programs for cerebral palsy be adapted, albeit with variations, to CZS.

Minor comments:

  1. On page 2, in the paragraph starting, “A number of family-based support programs have been established for families of children with neurodevelopmental disability in low- and middle-income contexts to respond to these unmet needs. One such program is ‘Getting to Know Cerebral Palsy’ (GTKCP), developed to educate and empower caregivers in the care of their child”, references for support programs are lacking. A few references here can strengthen the statement.
  2. The sample size used for PedsQL analysis is quite disparate (155 mothers of children with CZS versus only 47 mothers of unaffected children) The original Social and Economic Impacts study aims to recruit a total of at least 200 cases and 200 controls but there is no record of any numbers that were recruited. A reference for the case control study can justify the sample size used for statistical analysis.
  3. Among the limited number of facilities visited, how were the three sites mentioned in the article selected?  
  4. The scoping and identification of relevant articles fitting the inclusion (and/or exclusion) criteria in a systematic review is usually conducted by at least two independent researchers to ensure no bias and to make sure none of the relevant articles are missed. It was reported that only one researcher undertook this task and the reason was not explained.
  5. While the authors state that the healthcare providers agree that the concept of GTKCP can be applied to CZS, they really did not address their question, whether the similarities between CZS and cerebral palsy are sufficient to apply GTKCP to the current context.

In conclusion, the authors addressed their main question and supported the need for educational and support groups in a post-Zika context while stating their limitations, as well. By addressing the above comments, they can strengthen their argument.

Author Response

Dear reviewer   Many thanks for your review and comments.  We have considered your feedback and comments and have made the following adjustments to the manuscript.:   Comment 1: On page 2, in the paragraph starting, “A number of family-based support programs have been established for families of children with neurodevelopmental disability in low- and middle-income contexts to respond to these unmet needs. One such program is ‘Getting to Know Cerebral Palsy’ (GTKCP), developed to educate and empower caregivers in the care of their child”, references for support programs are lacking. A few references here can strengthen the statement.  Response: References to two other programmes: Hambisela (on which the GTKCP was originally based) and WHO’s Caregiver Skills Training Programme were added.
Comment 2: The sample size used for PedsQL analysis is quite disparate (155 mothers of children with CZS versus only 47 mothers of unaffected children) The original Social and Economic Impacts study aims to recruit a total of at least 200 cases and 200 controls but there is no record of any numbers that were recruited. A reference for the case control study can justify the sample size used for statistical analysis.  Response:The protocol changed part way through the study - initially we planned to collect data on cases and controls, but Recife found it wasn't feasible so Rio only collected the data on some controls and not all cases were sampled. We have added a note to this effect in the limitations section.   Comment 3: Among the limited number of facilities visited, how were the three sites mentioned in the article selected?  Response;The sites were selected by our Brazil partners and covered three areas that had seen significant numbers of Zika cases: Rio de Janeiro, Salvador and Recife. This enabled us to dialogue directly with physicians and other health professionals working directly with CZS cases.   Comment 4:The scoping and identification of relevant articles fitting the inclusion (and/or exclusion) criteria in a systematic review is usually conducted by at least two independent researchers to ensure no bias and to make sure none of the relevant articles are missed. It was reported that only one researcher undertook this task and the reason was not explained.  Response: The review was undertaken as a Masters project therefore the lead researcher worked individually, but under the supervision of one of the other researcher.  This has been better reflected in the methods and a sentence added in the limitations section to recognize this.                                                                                                                          Comment 5: While the authors state that the healthcare providers agree that the concept of GTKCP can be applied to CZS, they really did not address their question, whether the similarities between CZS and cerebral palsy are sufficient to apply GTKCP to the current context.  Response: We have redeveloped the discussion to better reflect responses to the questions posed in the introduction including some additional references. This hopefully better connects the CZS/CP linkage.